# Formation Control Technology of Fixed-Wing UAV Swarm Based on Distributed Ad Hoc Network

**Wenbo Suo [1], Mengyang Wang [2], Dong Zhang [2,*], Zhongjun Qu [1] and Lei Yu [3]**

[1] AVIC Xi'an Flight Automatic Control Research Institute, Xi'an 710076, China; suowenbo@mail.nwpu.edu.cn (W.S.); C919-zjqu@facri.com (Z.Q.)

[2] School of Astronautics, Northwestern Polytechnical University, Xi'an 710072, China; wangmengyang@mail.nwpu.edu.cn

[3] Science and Technology on Complex System Control and Intelligent Agent Cooperation Laboratory, Beijing 100074, China; jackzhangd@163.com

[*] Correspondence: zhangdong@nwpu.edu.cn; Tel.: +86-15891390256

**Abstract:** The formation control technology of the unmanned aerial vehicle (UAV) swarm is a current research hotspot, and formation switching and formation obstacle avoidance are vital technologies. Aiming at the problem of formation control of fixed-wing UAVs in distributed ad hoc networks, this paper proposed a route-based formation switching and obstacle avoidance method. First, the consistency theory was used to design the UAV swarm formation control protocol. According to the agreement, the self-organized UAV swarm could obtain the formation waypoint according to the current position information, and then follow the corresponding rules to design the waypoint to fly around and arrive at the formation waypoint at the same time to achieve formation switching. Secondly, the formation of the obstacle avoidance channel was obtained by combining the geometric method and an intelligent path search algorithm. Then, the UAV swarm was divided into multiple smaller formations to achieve the formation obstacle avoidance. Finally, the abnormal conditions during the flight were handled. The simulation results showed that the formation control technology based on distributed ad hoc network was reliable and straightforward, easy to implement, robust in versatility, and helpful to deal with the communication anomalies and flight anomalies with variable topology.

**Keywords:** fixed-wing UAV; UAV swarm formation; distributed ad hoc network; consistency theory; formation obstacle avoidance

## 1. Introduction

The fixed-wing UAV swarm has essential application prospects and has become a current research hotspot. Completing combat missions such as coordinated reconnaissance, early warning, strike, and evaluation in the military field; and realizing disaster emergency, geological survey, and pesticide-spraying tasks in the civilian field [1–4]. Formation control is one of the key issues to achieve UAV swarm flight [5,6]. Its primary con-tents include formation maintenance, formation switching, formation obstacle avoidance, and exception handling. The distributed wireless ad hoc network [7] is the core of realizing the cluster unmanned system [8]. The formation control based on the distributed ad hoc network can better reflect the distributed, networked, and centerless characteristics of the cluster system, which is the future development trend of cluster control.

The formation control technology for UAV swarm behavior has been extensively studied. Wang [9] proposed the leader–follower method. Its basic idea is that other UAVs follow a leader UAV as followers. Luo et al. [10] and Gu et al. [11] also adopted the leader–follower method to design a control method for the leader and followers in the formation. CamPa et al. [12], based on the leader–follower method, proposed a virtual leader method. Its main idea was to treat the formation of a multi-UAV formation as a rigid virtual structure.

When the formation moved as a whole, the UAV only needed to track the movement of the fixed point corresponding to the rigid body. Li and Liu [13] designed a synchronized position tracking controller to improve the effectiveness of using a virtual structure method to maintain formation geometry. Yun and Albayrak [14] applied behavior-control methods to study the formation of multiplatform formations, such as linear formation and circular formation. Chen and Luh [15] applied behavior-control methods to achieve the purpose of object transportation. Ginlietti et al. [16] used behavior methods to define the concept of the formation geometric center. When flying in formation, each aircraft needs to maintain a prescribed distance from the geometric center, and be able to perceive the movements of other aircraft and reconstruct the formation, similar to the behavior of migratory birds in nature. Joongbo et al. [17] proposed a feedback linearization method based on consistency for multi-UAV systems to maintain a specific time-varying formation flight geometry. Glavas et al. [18] applied the consistency research strategy to study the situation when there were random communication noise and information packet loss constraints in the network, and used the polygon method based on information exchange to achieve formation control. Yasuhiro and Toru [19] studied the cooperative control problem of multi-UAV systems, and proposed a cooperative formation control strategy with collision-avoidance capability using decentralized model predictive control (MPC) and consensus-based control. Zhao et al. [20] studied the problem of formation control of multiaircraft formation with time-varying formation characteristics when there was a spanning tree in the network topology based on the consistency theory, and obtained the stability conditions of the system. Dong et al. [21] designed a distributed formation controller based on the consistency theory, proving that as long as the network topology was guaranteed to have directional strong connectivity, even if the aircraft was lost during the formation flight, the multiaircraft system could still achieve stable formation control. Seo et al. [22] designed a consistent control protocol for the situation in which the network topology had fixed connectivity, and studied the problem of cooperative formation control of multiaircraft systems forming geometric formations. Tang et al. [23] used evolutionary control theory to complete distributed collaborative control of UAV formations. He and Lu [24] proposed a decentralized design method based on a UAV distributed-formation maintenance controller, decomposing the UAV formation model into decoupled parts and associated parts using robust control methods, and improved the distributed control method of the associated system designs of the controller to control the UAV formation flying. However, most of the research content was based on the formation controller design coupled with the UAV's underlying control system. It was assumed that the UAV had a three-channel autopilot and had the ability of instantaneous response [25]. These assumptions made it difficult to apply the formation algorithm to an actual UAV swarm control. Furthermore, the research content was insufficient in the control algorithm of real flight environments such as network topology jump, communication delay, and even weak communication.

Based on the consistency theory, this paper proposes a method for formation switching and obstacle avoidance based on waypoint planning for the problem of formation control of a fixed-wing UAV swarm in a distributed ad hoc networks. The organizational structure of the paper is as follows. The first part gives a general description of the problem of a fixed-wing UAV swarm formation in a distributed ad hoc network; the second part proposes a method for switching the formation of the UAV swarm based on the consistency theory; the third part designs a UAV swarm formation obstacle avoidance algorithm; the fourth part deals with the problems of flight abnormality and communication abnormality of the UAV swarm during the flight; the fifth part simulates and verifies the formation switching of the UAV swarm, the formation obstacle avoidance, and handling of anomalies during the flight; the sixth part analyzes and discusses the results; and finally, the seventh part summarizes the article.

## 2. Problem Formulation

This paper focuses on two critical problems of formation switching and formation obstacle avoidance in a distributed ad hoc network for UAV swarm formation control.

**Definition 1.** *Formation switch in distributed ad hoc network.*

For $n$ UAV, given an initial position $\mathbf{X}_i(0)$ of $\text{UAV}_i$, a UAV swarm forms a communication topology in the ad hoc network (as shown in Figure 1). Plan the waypoint $P_i = \{P_{i1}, \cdots, P_{ik}, \cdots\}$ of the $\text{UAV}_i$ under the dynamic constraints of the maximum turning angle constraint $\beta_{\max}$ and minimum route length constraint $d_s$, so that the distance between $\text{UAV}_i$ and other UAVs reaches the expected value $\Delta\mathbf{X}_{jiref}$ within the time $T$:

$$\left|\mathbf{X}_j(t) - \mathbf{X}_i(t) - \Delta\mathbf{X}_{jiref}\right| \to 0 \quad i = 1, 2, \cdots, n \tag{1}$$

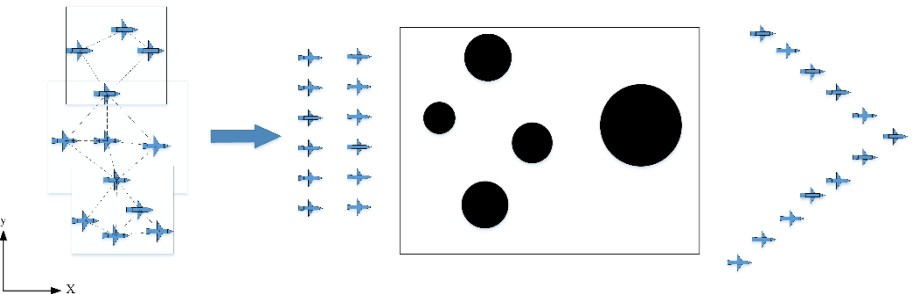

**Figure 1.** UAV swarm formation problem.

**Definition 2.** *Formation obstacle avoidance in distributed ad hoc network.*

For $n$ UAV, given an initial position X of $\text{UAV}_i$, intelligently split the UAV swarm into $N_c$ formation; the number of UAVs in the $p$th subformation is $N_p$. Plan the waypoint $P_i = \{P_{i1}, \cdots, P_{ik}, \cdots\}$ of the $\text{UAV}_i$ under the dynamic constraints of the maximum turning angle $\beta_{\max}$ and the minimum direct flying distance $d_s$, so that the UAVs will not collide through the rectangular obstacle avoidance area $S$ containing the circular obstacle $O$, and the distance between $\text{UAV}_i$ the and other UAVs will eventually reach the expected value $\Delta\mathbf{X}_{jiref}$; namely, reconstructed into the required formation.

Where $n = \sum\limits_{p=1}^{N_c} N_p$, $O$ is a collection of obstacles, and $O = \{o_1, o_2, \cdots, o_{oN}\}$, $oN$ is the number of obstacles, each obstacle $o_m$ can be described as a dyadic array $< R_{om}, R_m >$; $R_{om}$ is the center point of the $m$th circle; $R_m$ is the radius of the $m$th circle; $S$ is the rectangular obstacle avoidance area described as a ternary array $< S_o, L, W >$, where $S_o$ is the center point of the rectangular obstacle avoidance area; $L$ is the length of the rectangle; and $W$ is the width of the rectangle.

A reasonable formation-switching control method requires that the UAV can perform online formation switching based on the neighboring UAV information, and can meet the UAV dynamic constraints and can handle communication delays and flight anomalies, eliminating track deviation. Formation obstacle avoidance requires that a UAV swarm can effectively avoid obstacles, and can form a desired formation after the avoidance is completed. The UAV studied in this paper was a highly dynamic fixed-wing UAV with a uniform speed. The dynamic constraints were required to satisfy the minimum turning radius constraints and minimum track length constraints. The research space of this paper was the two-dimensional Euclidean horizontal plane.

## 3. Waypoint-Based Formation-Switching Method

The UAV studied in this paper was a highly dynamic fixed-wing UAV. Due to its high speed, it was difficult for the control system to update flight parameters in real time, and the errors were relatively large. Therefore, this article mainly considered the waypoint-based formation switching, and the online design of a small number of waypoints to complete the flight process of the UAV swarm formation, without the need to participate and change the design of the aircraft control system.

### 3.1. Consensus-Based Design for UAV Swarm Formation Control Protocol

The algebraic graph theory was used to describe the UAV swarm system and its behavior. Assume that the ad hoc network UAV swarm system has $n$ UAVs, and each UAV is regarded as a node, then the communication relationship is seen as an edge. An undirected graph $G = (V, E, \mathbf{A})$ represents the UAV swarm system, where $V = \{s_1, s_2, \cdots, s_n\}$ is a collection of nodes, $E = \{(s_i, s_j) \in s \times s, i \neq j\}$ is a collection of edges, and $\mathbf{A} = [a_{ij}]_{n \times n}$ represents an adjacency matrix with weights. The edge of the graph is indicated by $e_{ij} = (s_i, s_j)$. For an undirected graph, UAV$_i$ and UAV$_j$ can receive the information sent by each other, namely $(s_i, s_j) \in E \Leftrightarrow (s_j, s_i) \in E$. The adjacency matrix is defined as $a_{ii} = 0$, and $a_{ij} = a_{ji} \geq 0$, when $e_{ij} \in E$, $a_{ij} > 0$. The neighbor set of the node $s_i$ is defined as $N_i = \{s_i \in V | (s_i, s_j) \in E\}$.

An undirected graph $G_n$ was used to describe the communication topology relationship in the UAV swarm. At each moment $t$, the communication connection between the ad hoc network and UAV swarm forms a communication topology. For the vertex $i$ of graph $G$, let $x_i(t) \in R^q$ and $u_i(t) \in R^q$ denote the state variable and state information input variable of the UAV swarm, respectively, at the time $t$. The classic first-order continuous-time consistency protocol [26] is:

$$\dot{x}_i(t) = u_i(t) \quad i = 1, 2, \cdots, n \tag{2}$$

$$u_i(t) = -\sum_{j=1}^{n} a_{ij}(t)\big(x_i(t) - x_j(t)\big) \tag{3}$$

For any UAV$_i$, the initial state is $x(0) \in R^P$, when $t \to \infty$, there is $|x_i(t) - x_j(t)| \to 0$, which is called the state of UAV swarm system reaching consensus.

In this paper, the consensus algorithm needed to be applied to the formation switching of the UAV swarm. The distance between UAVs needed to reach the expected value eventually. Therefore, to improve the classic first-order consistency protocol, the first-order consistency protocol with reference location information was proposed, and the specific form was as follows:

$$\dot{x}_i(t) = u_i(t) = \frac{\sum\limits_{j=1}^{m} \left(a_{ij}(t)\left[(\mathbf{X}_j(t) - \mathbf{X}_i(t)) - \mathbf{X}_{jiref}\right]\right)}{\sum\limits_{j=1}^{m} a_{ij}(t)} i = 1, 2, \cdots, n \tag{4}$$

where $\mathbf{X}_{jiref}$ is the expected formation relative distance between UAV$_j$ and UAV$_i$.

### 3.2. Consensus-Based Waypoints Planning

3.2.1. Formation Waypoints

The UAV speed studied in this paper was constant, and the UAV swarm formation was controlled based on the route. Therefore, the consistency control protocol in Equation (4) needed to be discretized. First, the communication time of the UAV swarm was discretized, then the position status of each UAV was updated in real time according to the difference equation, and the discrete consistency control protocol can be given by:

$$\mathbf{X}_i[k+1] = \mathbf{X}_i[k] + \Delta\mathbf{X}_i[k] + \mathbf{D} \tag{5}$$

$$\Delta \mathbf{X}_i[k] = \frac{\sum\limits_{j=1}^{m} \left( a_{ij}[k]((\mathbf{X}_j[k] - \mathbf{X}_i[k]) - \mathbf{X}_{jiref}) \right)}{\sum\limits_{j=1}^{m} a_{ij}[k]} \qquad (6)$$

where $k$ represents a communication event, which is a formation process of the UAV swarm formation waypoint; $\mathbf{D}$ is the distance required to switch the formation, and is selected according to actual needs; $\mathbf{X}_j[k]$ and $\mathbf{X}_i[k]$ are the position status of the UAV$_j$ and UAV$_i$ at time $k$, also called formation waypoint; $a_{ij}[k] \in R \times R$ is the adjacency weight matrix of the communication topology, and its elements are defined as:

$$a_{ij} = \begin{cases} 1 & (v_j, v_i) \in E \\ 0 & (v_j, v_i) \notin E \end{cases} \qquad (7)$$

The undirected graph in this article did not allow self-loops, so $a_{ij} = 0$.

During the flight, UAV$_i$ established a communication connection with other UAVs in the ad hoc network, forming a formation waypoint $\mathbf{X}_i[k]$ within each communication event $k$ according to Equations (5) and (6).

According to the consistency theory, it can be proved that for any UAV$_i$ with an initial value $\mathbf{X}_i[0] \in R^P$, the time-varying communication topology union is fully connected in a formation transformation [27], namely $\sum\limits_{k=0}^{Nk} a_{ij}[k] > 0 \ (i \neq j)$, when $k \to Nk$, there is $\left| \Delta \mathbf{X}_{ji}[k] - \Delta \mathbf{X}_{jiref}[k] \right| \to 0$; that is, the relative distance between the UAV swarm reached the desired value, and the formation switch was realized. If the initial state of the communication topology $a_{ij}[0]$ was full connectivity; namely, the communication was in the normal state, the formation fly could be achieved as the expected formation when $k = 0$.

### 3.2.2. Flying to Formation Waypoint in Consistent Time

Once the formation waypoints were obtained, only local waypoints from the current position of the UAV$_i$ to its formation waypoint needed to be designed so that the flight time of each UAV was equal. In this section, a distributed UAV swarm flying method based on time consistency is proposed, and local waypoints are designed under dynamic constraints to achieve UAV swarm formation switching when flying at a constant speed.

(1) Dynamic constraints

Maximum turning angle constraint: when the UAV turns according to the waypoint, the turning angle $\beta$ (as shown in Figure 2) needs to be lower than the maximum turning angle $\beta_{\max}$:

$$\beta \leq \beta_{\max} \qquad (8)$$

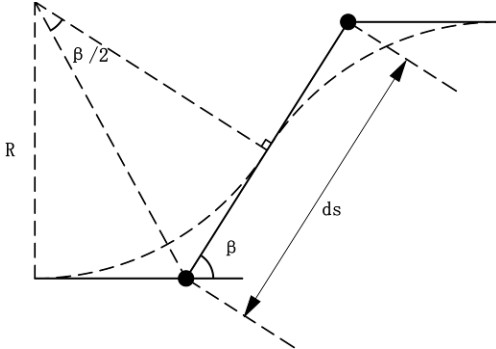

**Figure 2.** Dynamic constraints.

Minimum route length constraint [28]: assuming that the UAV was flying under the available overload at the turn, the turning radius $R$ was a fixed value. As shown in Figure 2,

the turning angle of the UAV in this section of the route was $\beta$. For the UAV to make two consecutive turns, the route of this section needed to be lower than the minimum track length constraint $d_s$:

$$d_s = 2 \times R \times \tan \frac{\beta}{2} \tag{9}$$

(2)　Waypoint design

When a UAV flew to formation waypoint, we first took the UAV that was going to fly the longest path as the time base, and then planned for other UAVs to fly around in the horizontal plane so that the final time to reach the formation waypoint was consistent.

The longest path $C_{\max}$ is:

$$C_{\max} = \max(C_i) + d_y \tag{10}$$

where $d_y$ is the vertical flight distance margin to ensure that the vertical distance meets the constraint of the minimum route length $d_s$. The turning angle of this method is a fixed value $\beta = 90°$, so we set $d_y = 2 \times d_s = 4 \times R$, $C_i$ as the route distance that the UAV$_i$ needed to fly:

$$C_i = \Delta x_i + \Delta y_i + \Delta t_i \times V \tag{11}$$

where $\Delta x_i$ and $\Delta y_i$ are the horizontal distance and vertical distance of UAV$_i$ from the current position to the formation waypoint; $V$ is the flight speed of the UAV; and $\Delta t_i$ is the waiting time of UAV$_i$, which can be selected according to the actual project. If it was a formation switch scenario, $\Delta t_i = 0$. If it was a formation assembly scenario, $\Delta t_i$ was the launch interval of UAV$_i$ from the first UAV.

The dynamic constraints of the flight plan needed to consider the maximum turning angle constraint and the minimum route length constraint. The flight project was divided into the following sections:

As shown in Figure 3, the flight project was composed of four right-angle turning sections. The solid line is the planned route, which was sections ①, ②, ③, ④, and ⑤. The dashed line is the actual flight route considering the UAV turning process. The solid point is the UAV waypoint. To make the flying distance the same, the length of the distances of ①, ②, ③, ④, and ⑤ were designed as:

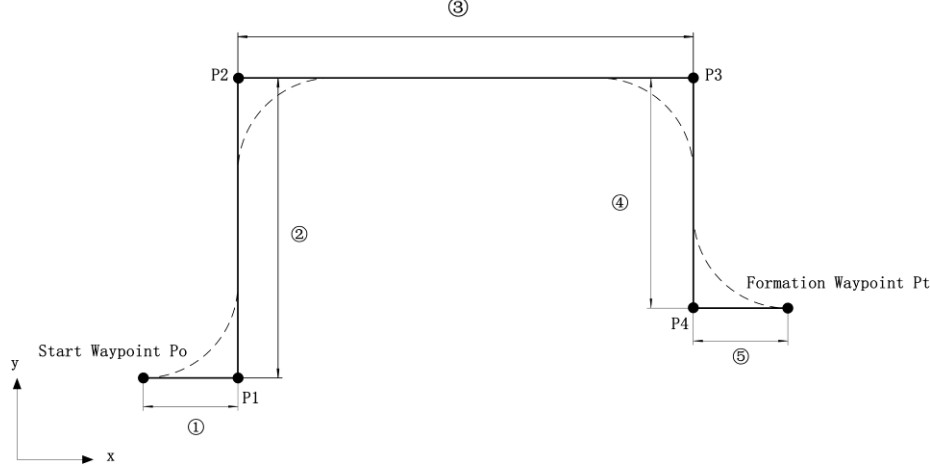

**Figure 3.** Dynamic constraints.

$$\begin{cases} L_{1x,i} = L_{5x,i} = r \\ L_{3x,i} = \Delta x_i - 4 \times r \\ L_{2y,i} = L_{4y,i} = \frac{C_{\max} - \Delta x_i - \Delta y_i}{2} \end{cases} \tag{12}$$

where $L_{1x,i}$, $L_{3x,i}$, and $L_{5x,i}$ are the horizontal distances of UAV$_i$ in segments ①, ③, ⑤; and $L_{2y,i}$, $L_{4y,i}$ are the vertical distances of UAV$_i$ in segments ② and ④, respectively.

The local waypoints of the UAV$_i$ could be calculated according to the starting waypoint and the distance lengths of ①, ②, ③, ④, and ⑤:

$$\begin{cases} P_{i1}(x_{i1}, y_{i1}) = P_{i0}(x_{i0} + L_{1x,i}, y_{i0}) \\ P_{i2}(x_{i2}, y_{i2}) = P_{i1}(x_{i1}, y_{i1} + L_{2y,i}) \\ P_{i3}(x_{i3}, y_{i3}) = P_{i2}(x_{i2} + L_{3x,i}, y_{i2}) \\ P_{i4}(x_{i4}, y_{i4}) = P_{i3}(x_{i3}, y_{i3} - L_{4y,i}) \end{cases} \tag{13}$$

At this point, all local route points could be obtained. The UAVs could arrive at the formation waypoints at the same time, thus forming the expected formation.

In this section, a waypoint-based distributed ad hoc network formation-switching control method, derived from the consistency theory, only needed to obtain the formation waypoint from the position information of the UAVs, and then plan the local waypoint from the current waypoint to the formation waypoint under dynamic constraints. The UAV only needed to reach the online planned waypoint under the control of its flight control system. It did not need to call the UAV's control system to track flight parameters in real time, and only required the design of four local waypoints. This method has a small amount of calculation, is practical and straightforward, and is conducive to implementation in engineering. It can also realize dynamic formation control of UAVs when some communication networks are lost and the topology structure changes.

## 4. Waypoint-Based Formation Obstacle Avoidance Algorithm

The traditional formation obstacle avoidance method uses an artificial potential field method for obstacle avoidance or an intelligent algorithm to search for waypoints. Still, considering the real-time nature of obstacle avoidance and engineering applications, the artificial potential field method needs to participate in the design of the control system. When facing a high-dynamics UAV, it is challenging to update flight parameters in real time, as it presents significant errors and low reliability. Using the intelligent algorithm to plan trajectories, if high precision is required, planning the trajectory of a single UAV is still too slow, and if there are many planned waypoints, it cannot meet the dynamic constraints of the UAV.

In this paper, the intelligent path search algorithm was used to search for obstacle avoidance channels through which UAVs could pass, and the UAV swarm was divided into multiple smaller formations according to the number of UAVs that could pass through the avoidance channel.

### 4.1. Consensus-Based Design for UAV Swarm Formation Control Protocol

Obstacle-avoidance principles include the A* algorithm and the smallest enclosing convex polygon of a set of points (SECP) decision principle.

(1)    A* algorithm

The A* algorithm [29,30] is the most effective direct search method for solving the shortest path in a static road network. We only needed to search for the formation channel between the obstacle circles, and there was no need to search for high-precision track points, so we could use the traditional A* algorithm:

$$f(o) = g(o) + h(o) \tag{14}$$

where $f(n)$ is the cost estimate from the initial state to the target state via state $n$; $g(n)$ is the actual cost from the initial state to state $n$ in the state space; and $h(n)$ is the estimated cost of the best path from state to the target state. The shortest path could be determined by searching from the starting point according to the valuation function in Equation (14) to the ending point.

(2)    SECP determination method

Given a plane point set $A = \{(x_i, y_i) | x_i, y_i \in R\}$, we determined the smallest enclosing convex polygon of a set of points $A_i$ (SECP): we connected the points two by two to form a line segment set $S = \{(A_i, A_j) | A_i, A_j \in A, i \neq j\}$. If all the other line segments were on the side of the line where a line segment $S_{convex}$ was located, then the line segment $S_{convex}$ where this line (e.g., the dotted line in Figure 4) lay was a side of the required convex polygon SECP.

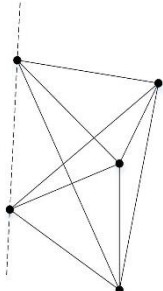

**Figure 4.** Determination method.

*4.2. Formation Obstacle Avoidance Algorithm*

To obtain multiple formation passage paths, firstly, it was necessary to determine the entry points and the exit points according to the SECP determination method; secondly, to determine the formation obstacle avoidance path in combination with the A* search method; and finally, to determine the obstacle avoidance waypoint of the UAV swarm. The algorithm flow is shown in Figure 5; the specific steps were as follows:

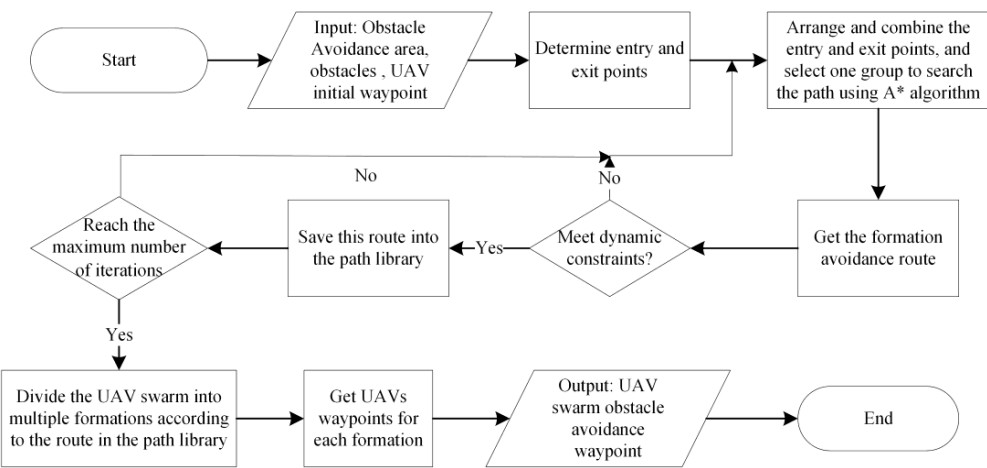

**Figure 5.** Principle flow chart of obstacle avoidance algorithm.

Step 1. Determine entry and exit points.

As shown in Figure 6, we connected the center points of circles two by two to get the connecting line set $CirSeg = \{CirSegment_k | k = 1, 2 \cdots, Nk\}$, and obtained the entry and exit line segments according to the SECP determination method, and further obtained the in-point set *EnPoint* and the out-point set *ExPoint*. The specific process of the algorithm was as follows Algorithm 1:

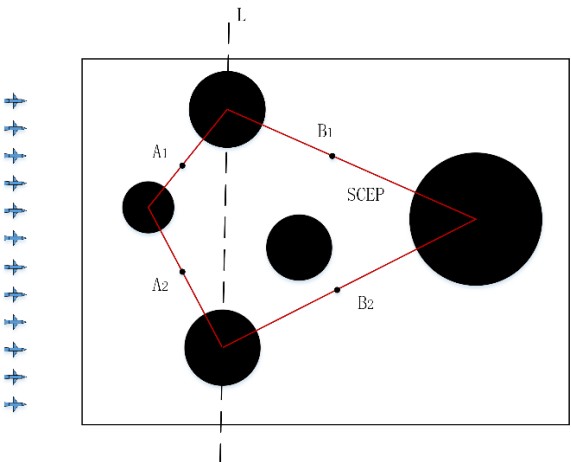

**Figure 6.** Determination of entry and exit points.

---

**Algorithm 1.** Get the entry points and exit points

---

**Input:** *O*: Obstacle collection *S*: Obstacle avoidance area
**Output:** $ExPoint = \left\{ ExitPoint_j \middle| j = 1, 2 \cdots \right\}$ $EnPoint = \left\{ EntryPoint_i | i = 1, 2 \cdots \right\}$
1: *CirSeg*← connect the two-point in $R_o = \{R_{om} | m = 1, \cdots, n\}$
2: *MidLine*← connect the maximum and minimum of point $R_{om}$ in *y*-direction
3: **for** $k = 1$ **to** *Nk* **do**
4:       $CirLine_k \leftarrow CirSegment_k$
5:       **if** $\{$CirSeg $- CirSegment_k\}$ in the same side of $CirLine_k$ **then**
6:             $CirSegment_k \in SCEP$ (e.g., SCEP in Figure 6)
7:             $PathSegement \leftarrow \{CirSegment_k - CirSegment_k \cap O\}$
8:             **if** *PathSegement* on the left side of *MidLine* **then**
9:                   $EnPoint \leftarrow$ the middle point of *PathSegement* (e.g., A in Figure 6)
10:            **else** $ExPoint \leftarrow$ the middle point of *PathSegement* (e.g., B in Figure 6)
11:            **end if**
12:      **end if**
13: **end for**
14: **return** *EnPoint*, *ExPoint*

---

Step 2. Determine formation avoidance path.

As shown in Figure 7, we combined the entry and exit points set *EnPoint ExPoint* from Algorithm 1 and the processed channel segment *CirSeg* (e.g., *M* in Figure 7) to find the obstacle avoidance path set *AvoidPath* with the A* method. The specific process of the algorithm was as follows Algorithm 2:

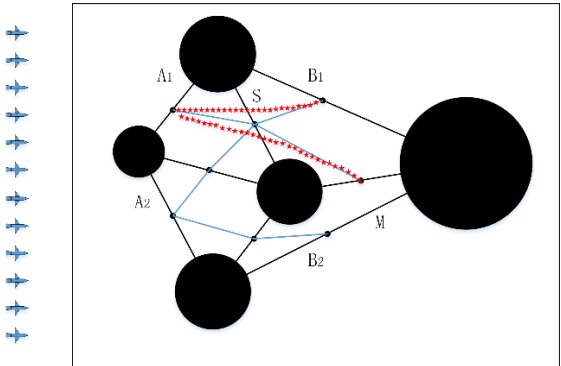

**Figure 7.** Determination of the formation avoidance path.

**Algorithm 2.** Get the avoidance path

---

**Input**: $O$, $S$, $ExPoint$, $EnPoint$

**Output**: $AvoidPath = \left\{ AvoidPath_{(i \to j),q} \middle| q = 1, 2, \cdots, Nq \right\}$

1: **for** $k = 1$ **to** $Nk$ **do**
2:     **if** $CirSegment_k \cap O$ **then** erase $CirSegment_k$
3:     **else if** $CirSegment_k \cap \{ \text{CirSeg} - CirSegment_k \}$ **then** erase $CirSegment_k$
4:     **end if**
5: **end for**
6: use $O$ and $S$ initialize A* map
7: **for** $i = 1$ **to** $Ni$ **do**
8:     **for** $j = 1$ **to** $Nj$ **do**
9:         search $Path_{i \to j}$ from $EntryPoint_i$ to $ExitPoint_j$ by A* (e.g., $S$ in Figure 7)
10:        **for** $k = 1$ **to** $Nk$ **do**
11:            **if** $Path_{i \to j} \cap CirSegment_k$ **then**
12:                $Pathsegement \leftarrow \{ Csegment_k - Csegment_k \cap O \}$
13:                $AvoidPath_{(i \to j),q} \leftarrow$ the middle point of $Pathsegement$
14:            **end if**
15:        **end for**
16:     **end for**
17: **end for**
18: **return** $AvoidPath$ (e.g., the blue line in Figure 7)

---

Step 3. Determining the UAV formation obstacle avoidance waypoint.

As shown in Figure 8, we extended $AvoidPath$ to the boundary of the obstacle avoidance area (e.g., $P$ in Figure 8), deleted the formation obstacle avoidance path that did not meet the UAV dynamic constraints, and calculated the number of UAVs that the path of the $i$th entry point could pass through:

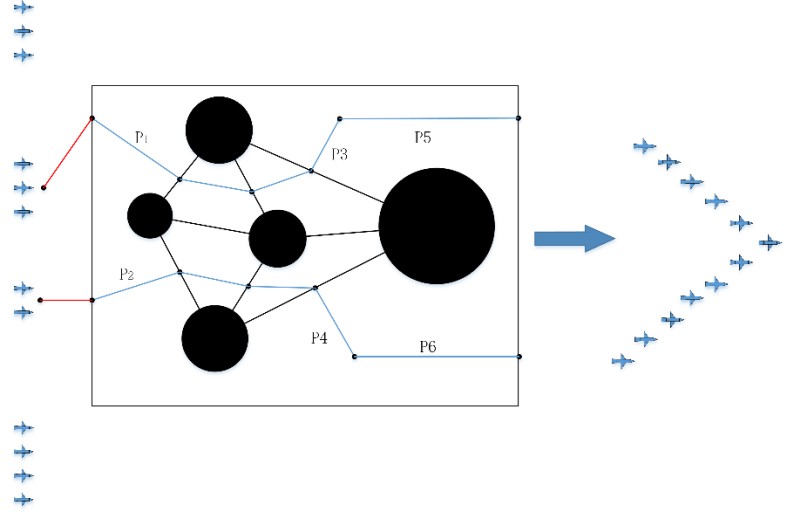

**Figure 8.** Determining the UAV formation obstacle avoidance waypoint.

$$n_{i \to j} = \min \left( \left\lfloor \omega \times \frac{L_{(i \to j),q} - 2 \times d_{safe}}{d} \right\rfloor + 1 \right) \, q = 1, 2, \cdots, Nq \qquad (15)$$

$$\begin{cases} n_i = \max n_{i \to j} \; j = 1, 2, \cdots, Nj \\ jmax = \operatorname{argmax}(n_i) \\ N_{pass} = \sum\limits_{i=1}^{Ni} n_i \end{cases} \qquad (16)$$

where $d$ is the vertical interval distance of the UAV, $d_{safe}$ is the safe distance of the UAV, $L_{(i \to j),q}$ is the length of the $q$th formation channel segment from the $i$th formation path to

*j*th formation path, $n_j$ is the number of UAV that can be passed by the *j*th formation path, $\omega$ is the scaling scale, and $0 < \omega \leq 1$ is to adjust the number of UAVs through the channels.

Then, we calculated the obstacle avoidance waypoint *PathPoint* of UAV$_i$ based on *AvoidPath* obtained from $n_i$ and Algorithm 2; the specific process was as follows Algorithm 3:

---

**Algorithm 3.** Get fly path point

---

**Input**: *O*, *S*, **X**(0), *AvoidPath*, *v*
**Output**: $PathPoint = \{PathPoint_{uavi}|uavi = 1, \cdots, N\}$
1: extent $AvoidPath_{i \to j}$ to *S* 's boundary (e.g., *P* in Figure 8)
2: **if** $AvoidPath_{i \to j}$ do not satisfy constraints $R_{res}$ and $d_s$ **then** erase
3: **end if**
4: calculate $n_i$ by Equation (15)
5:      $CirLine_k \leftarrow CirSegment_k$
6: split UAV swarm into $Ni$ sub-forms
7: $PathPoint \leftarrow$ calculate $PathPoint_{uavi}$ based $AvoidPath_{(i \to jmax),q}$
8: **if** $N_{pass} \leq N$ **then**
9: $PathPoint \leftarrow \{N - N_{pass}\}$ UAV fly around the obstacle area
10: **end if**
11: **for** $uavi = 1$ **to** $N$
12:      $T_{uavi} = |PathPoint_{uavi}|/v$
13: **end for**
14: $PathPoint \leftarrow$ use formation conversion algorithm with $T_{uavi}$ and *PathPoint*
15: **return** *PathPoint*

---

## 5. Handling Exceptions

This paper dealt with two kinds of abnormal situations: flight abnormity and communication abnormalities during the formation flight of distributed ad hoc network UAV swarm. The processing methods of each category were shown in Figure 9, and the detailed processing methods were shown below.

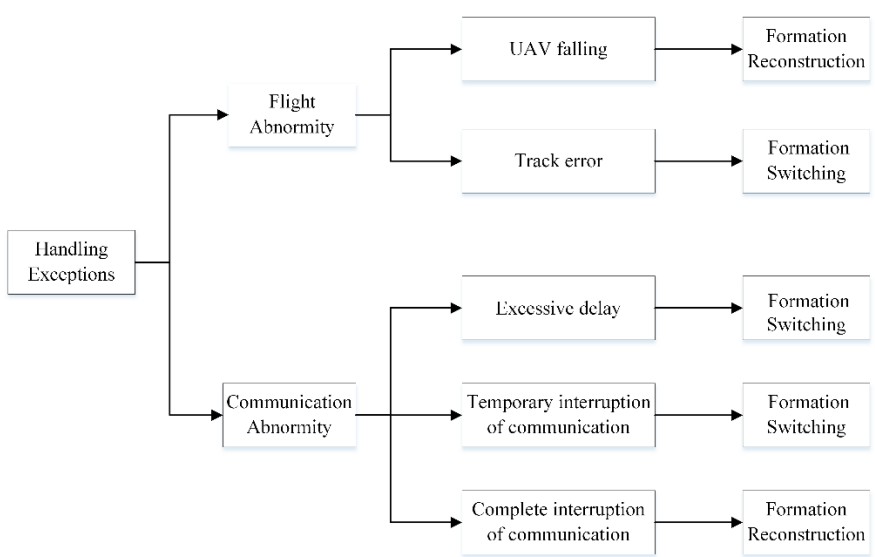

**Figure 9.** Handling exceptions.

### 5.1. Flight Abnormity

For the abnormal situation in which the tracking error was too large, the above-mentioned distributed ad hoc network online formation switch algorithm could be used for dynamic adjustment, and had the characteristics of not being related to the initial position of the UAV. In response to the falling of the UAV during the flight, the remaining

UAVs switched formation online using the formation switch algorithm; that is, formation reconstruction.

### 5.2. Communication Abnormalities

Communication abnormalities mainly considered three situations. First, the excessive communication delay included two cases, which were the entire UAV swarm's delay and some members' delay of the UAV swarm. Second, the communication was completely disconnected; that is, the other UAV communication links in the UAV swarm were completely disconnected and could not be restored. Third, the communication was partially interrupted; that is, the UAV communication link of other parts of the UAV swarm was temporarily disconnected, and it could be restored after some time.

If the communication of some UAVs was completely disconnected, this could be regarded as the situation of UAV formation to reconstruct the formation. If there was a delay in communication, the offset error could be eliminated according to the online formation-switching algorithm. If the communication of some UAVs was temporarily interrupted, multiple iterative formation flights could be performed based on the distributed ad hoc network online formation-switching algorithm.

## 6. Simulation Analysis

In this paper, 12 UAVs were used for formation assembly, switching, and formation flight simulation under abnormal flight and communication environments, and 8 UAVs were used for formation obstacle avoidance simulation. In the simulation experiment, the algorithm was programmed in the C++ language, the platform tool was Microsoft Visual Studio 2016, and the hardware environment was a PC with an inter-core i5-4210 CPU, 2.60 GHz dual-core processor, and 8 GB memory.

### 6.1. Formation Assembly and Formation Switching

The 12 UAVs took off in succession. After each UAV passed the assembly point A, the formation-switching algorithm began to take over the UAV, planning the trajectory of the UAV; the formation was assembled into an inverted V-shape; and then the formation-switching algorithm was enabled again to change the swarm into a V-shaped formation, as shown in Figure 10. The simulation parameters that had to be set are shown in Table 1. The simulation results of formation assembly and formation switching in the horizontal plane are shown in Figures 11 and 12.

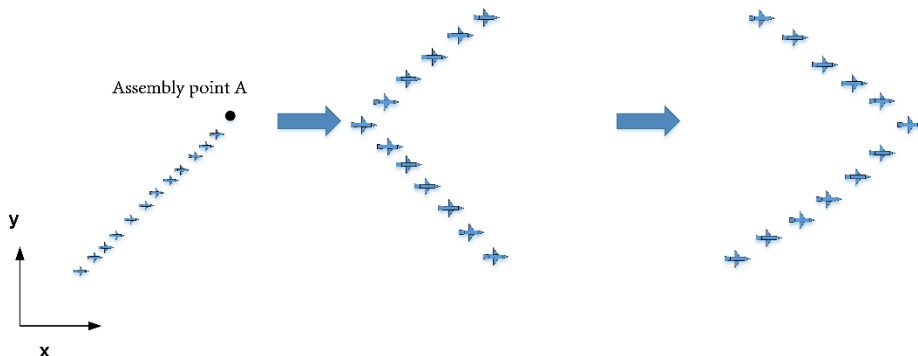

**Figure 10.** UAV swarm formation assembly and formation switching.

**Table 1.** Simulation parameters.

| UAV Swarm Attributes | Parameter Value |
|---|---|
| Number of UAVs $n$ | 12 |
| Launch interval | 5 s |
| UAV speed $V$ | 30 m/s |
| Maximum turning angle constraint $\beta_{max}$ | 90° |
| Turning radius $R$ | 300 m |
| Minimum track length constraint $d_s$ | 600 m |
| Assembly point A | (0 m, 2000 m) |
| Communication topology | Fully Connected |
| Assembled formation | Inverted V-Shape |
| Switch formation | V-Shape |
| Formation interval in x direction | 100 m |
| Half vertex angle of V-shaped | 45° |
| Formation-switching distance D | Adaptive (met the minimum switching distance) |

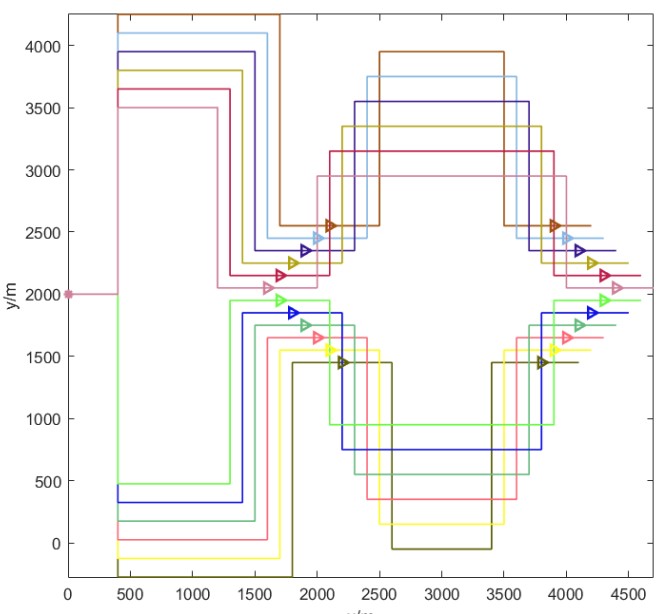

**Figure 11.** Formation assembly and formation switching.

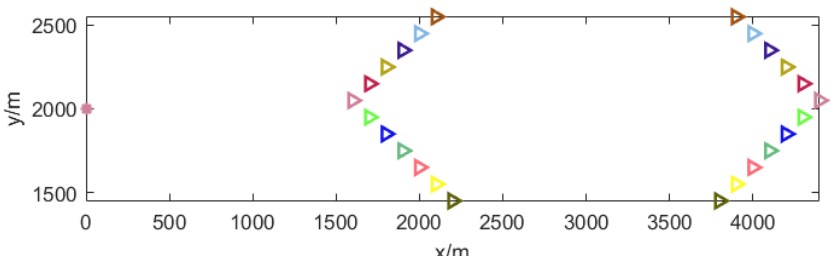

**Figure 12.** Assembly formation and switch formation.

The assembly formation was V-shaped, the formation switching was changed from V-shaped to column formation, the vertical formation interval in the y-direction was 100 m, and other parameters remain unchanged. The simulation is shown in Figures 13 and 14.

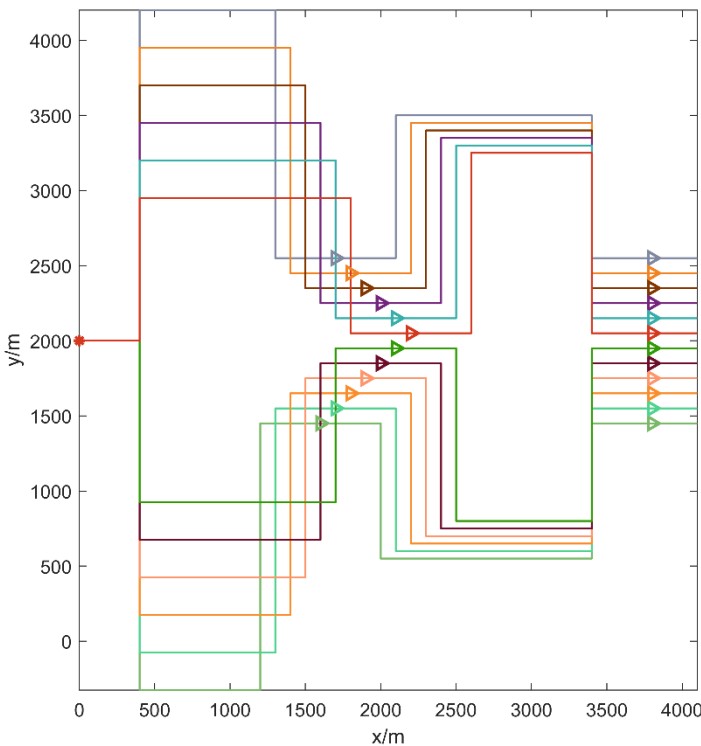

**Figure 13.** Formation assembly and formation switching—scene 2.

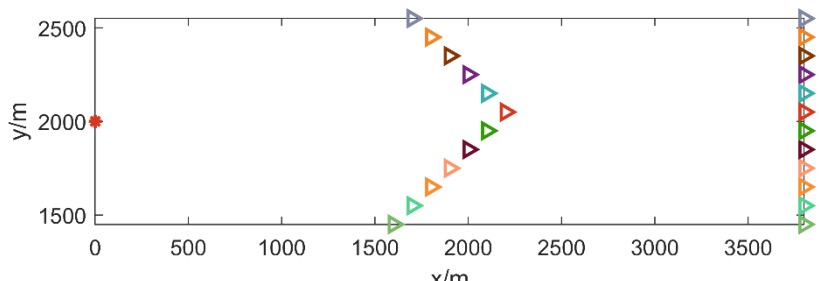

**Figure 14.** Assembly formation and switch formation—scene 2.

It can be seen in Figures 12 and 14 that when the communication topology of UAV swarm was fully connected, formation aggregation and formation switching could make the distance between each UAV meet the desired requirements after a formation interval correction; namely, the communication event was $k = 0$. If the communication topology was partially interrupted, the formation interval had to be corrected again. This method could simply and quickly converge to the final expected formation interval value. It can be seen in Figures 11 and 13 that this formation method could quickly plan the route to the formation waypoints under the constraints of the UAV dynamics, and has a high engineering application value.

### 6.2. Formation Obstacle Avoidance

The obstacle avoidance area was rectangular, and the relevant parameters are shown in Table 2. Eight UAVs formed a column formation to enter the obstacle avoidance area. The obstacle information is shown in Table 3. The initial position information of UAVs is shown in Table 4. The maximum turning angle constraint was 90°, and the minimum direct flight distance constraint was calculated according to Equation (9). The UAV swarm passed through the obstacle and was reconstructed into a V-shape. Other parameters of UAV swarm are shown in Table 1. The horizontal plane simulation results are shown in Figure 15.

**Table 2.** Obstacle avoidance area parameters.

| Obstacle Avoidance Area Properties | Parameter Value |
|---|---|
| Rectangle center point | (7000 m, 5000 m) |
| Rectangular area length | 10,000 m |
| Rectangular area width | 10,000 m |

**Table 3.** Obstacle parameters.

| Obstacle Index | X/m | Y/m | Radius/m |
|---|---|---|---|
| 1 | 3000 | 7000 | 500 |
| 2 | 4000 | 3000 | 500 |
| 3 | 6500 | 5000 | 500 |
| 4 | 11,000 | 3000 | 900 |
| 5 | 11,000 | 6000 | 600 |

**Table 4.** Initial position of UAV.

| UAV Index | X/m | Y/m |
|---|---|---|
| 1 | 1000 | 3000 |
| 2 | 1000 | 6500 |
| 3 | 1000 | 3500 |
| 4 | 1000 | 6000 |
| 5 | 1000 | 4000 |
| 6 | 1000 | 5500 |
| 7 | 1000 | 4500 |
| 8 | 1000 | 5000 |

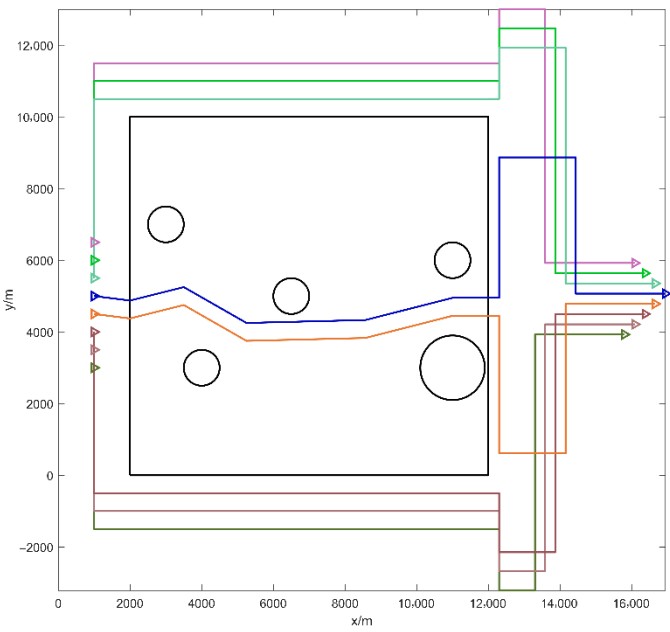

**Figure 15.** UAV formation obstacle avoidance.

We reset obstacles for simulation, and the obstacle information is shown in Table 5. Other parameters remain unchanged; the horizontal plane simulation results are shown in Figure 16.

**Table 5.** Obstacle parameters—scene 2.

| Obstacle Index | X/m | Y/m | Radius/m |
|:---:|:---:|:---:|:---:|
| 1 | 3000 | 5000 | 700 |
| 2 | 6000 | 14,000 | 1000 |
| 3 | 6500 | 5000 | 700 |
| 4 | 7000 | 8500 | 1300 |
| 5 | 11,000 | 4000 | 900 |

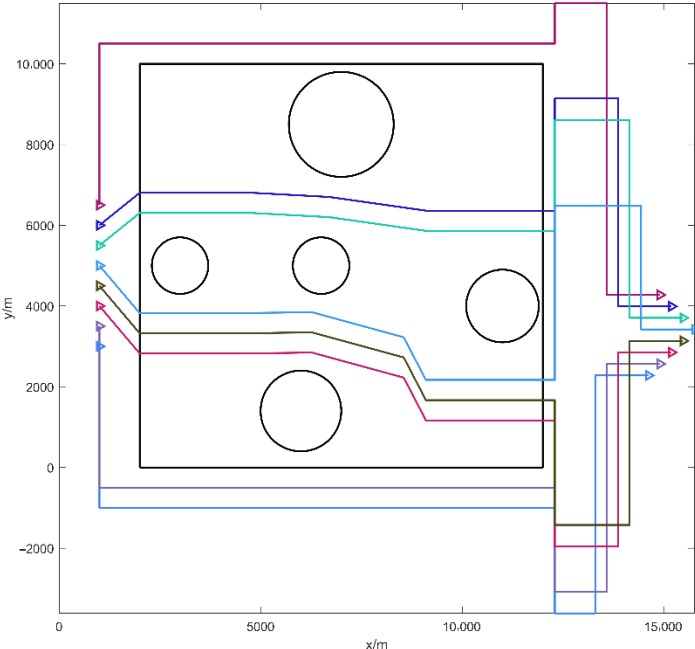

**Figure 16.** UAV formation obstacle avoidance—scene 2.

As can be seen in Figures 15 and 16, the UAV swarm could be intelligently divided into multiple smaller formations to pass obstacles or fly around to avoid obstacles. Finally, it could be reconstructed into the expected formation. The simulations showed that the algorithm could quickly and flexibly plan the cooperative obstacle avoidance path and had the ability of online formation to avoid obstacles.

### 6.3. Handle Exceptions

#### 6.3.1. Flight Abnormity

Suppose the number of UAV swarm was 12, and the assembly formation was an inverted V-shaped formation. After the formation assembly, four UAVs were randomly dropped or lost. Then, the remaining eight UAVs were reconstructed into a column formation. Other simulation parameters were still as shown in Table 1, and the simulation results are shown in Figures 17 and 18:

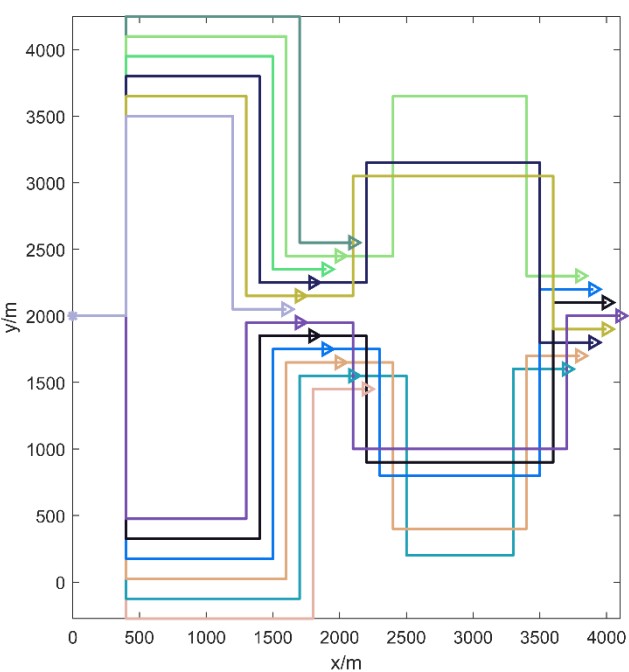

**Figure 17.** Formation reconstruction.

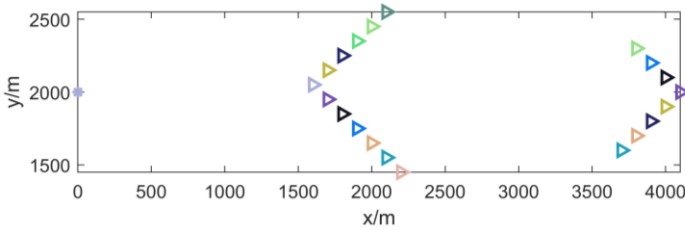

**Figure 18.** Formation reconstruction process.

It can be seen in Figures 17 and 18 that after the UAV flight abnormally fell, then the required formation could be reconstructed according to this plan.

### 6.3.2. Communication Abnormity

In actual flight, the communication topology may be temporarily interrupted. At this time, a communication event, $k = 0$, is insufficient to meet the expected formation, and formation interval distance correction is required again.

In this simulation, a formation switch was performed from an inverted V-shape to a V-shape. It was assumed that in the initial communication event, namely $k = 0$, the communication failure rate reached 18%, and in the second communication event, namely $k = 1$, the communication failure rate reached 9%. Other parameters were as shown in Table 1, and the results are shown in Figures 19 and 20.

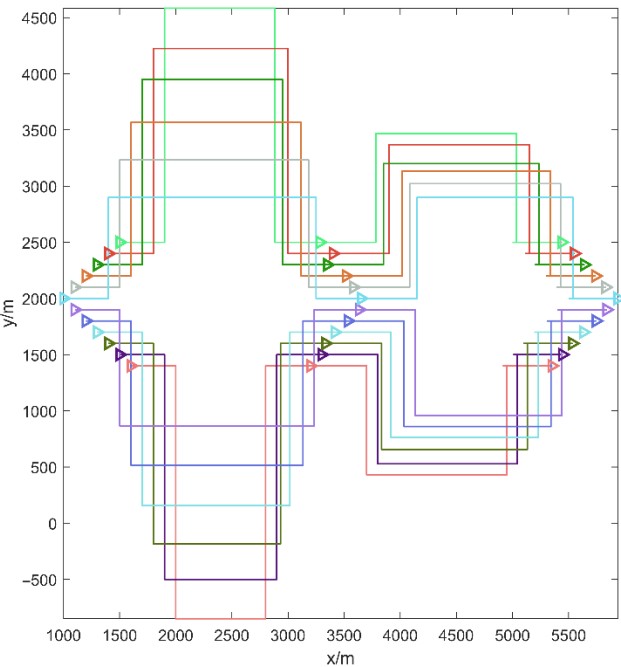

**Figure 19.** Formation switch under abnormal communication.

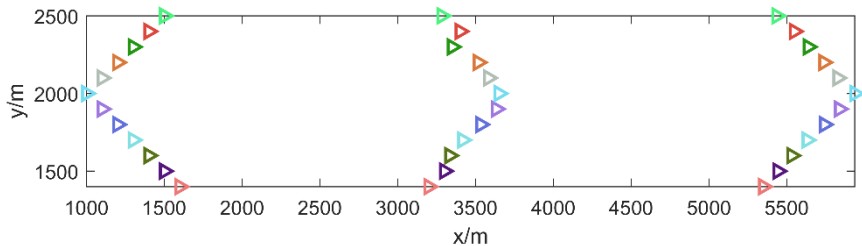

**Figure 20.** Formation change process under abnormal communication.

It can be seen in Figures 19 and 20 that in the case in which some UAVs and other UAVs were lost in the communication network, the first formation interval distance correction could not obtain the desired formation. In the second correction, the communication network still had local faults, but eventually formed the expected formation. So, the formation-switching algorithm was effective in dealing with communication abnormity.

### 6.4. Simulation Comparison

6.4.1. Formation-Switching Method

The formation-switching control method proposed in this paper mainly dealt with the communication abnormalities of the UAVs during the flight. Therefore, the classic consistency control method [31], which could handle changes in the communication topology, was selected to compare to illustrate the effectiveness of the algorithm.

We selected six UAVs for simulation, and used the simulation scenario in Section 6.3.2 to set the consistency control step to meet the minimum algorithm stability requirement, which was 0.2 s. The simulation of the classical consistency control method is shown in Figure 21, and the method proposed in this paper is shown in Figure 22.

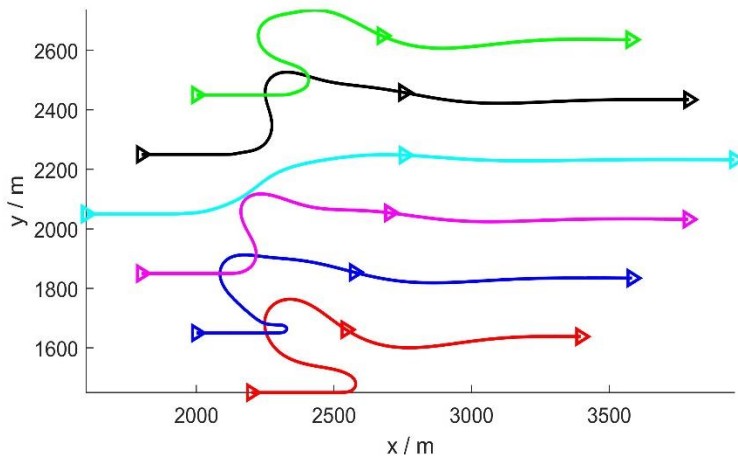

**Figure 21.** Formation switching of the classical consistency control method.

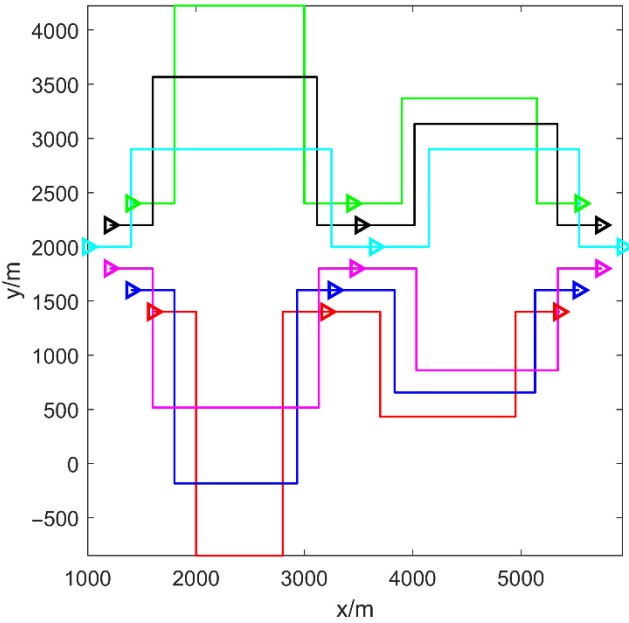

**Figure 22.** Formation switching.

The algorithm performance during the simulation process was recorded, as shown in Table 6.

**Table 6.** Simulation comparison.

| Method | Classical Consistency Control Method | Proposed Method |
|---|---|---|
| Number of communication requests | 200 times | 2 times |
| Average solution time per communication | 125 ms | 53 ms |
| Total running time of the algorithm | 1521 ms | 131 ms |

The simulation results showed that when the step length was 0.5 s, in the case of abnormal communication, the classic consistent formation control method needed to communicate with neighboring UAVs 200 times before the formation could be changed from an inverted V to a V-shape. The route-based formation-switching method proposed in this paper only needed two communications to obtain the required formation waypoints, which greatly reduced the pressure on the airborne communication system.

In addition, the classic consensus algorithm obtained the coordinated route of the UAV by controlling the deflection angle of the track. The average solution time per communication took a longer time, and required high control system performance, which could not guarantee the real-time performance of online formation flying. The solution proposed in this paper was time-consuming, and only required the UAV to perform direct flight and turning maneuvers. It was easy to control during the flight and easy to implement in engineering. However, the disadvantage was that the planned path distance was relatively long.

6.4.2. Formation Obstacle Avoidance Method

The sparse A * algorithm [30] was selected and compared with the formation obstacle avoidance algorithm proposed in this paper to illustrate the effectiveness of the algorithm. The simulation scenario used scenario 2 as given in Section 6.2. The sparse A* simulation results are shown in Figure 23.

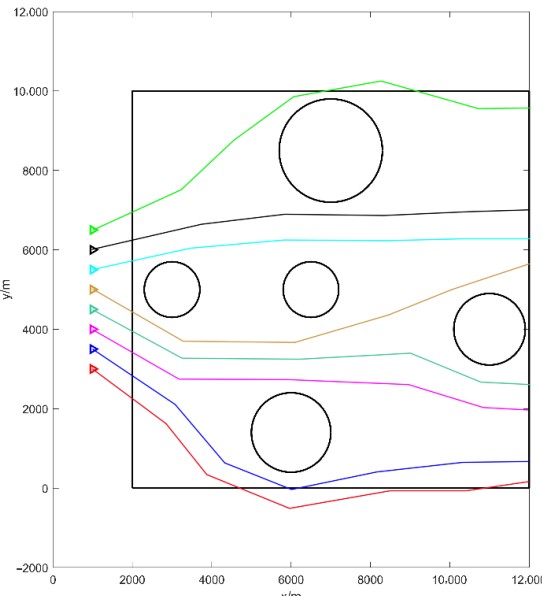

**Figure 23.** The sparse A * algorithm for formation obstacle avoidance.

In order to analyze the complexity of the two algorithms, the number of UAVs in the formation was increased successively, and the two algorithms were used to simulate the formation obstacle avoidance simulation. The relationship between the algorithm running time and the number of drones was obtained as shown in Figure 24.

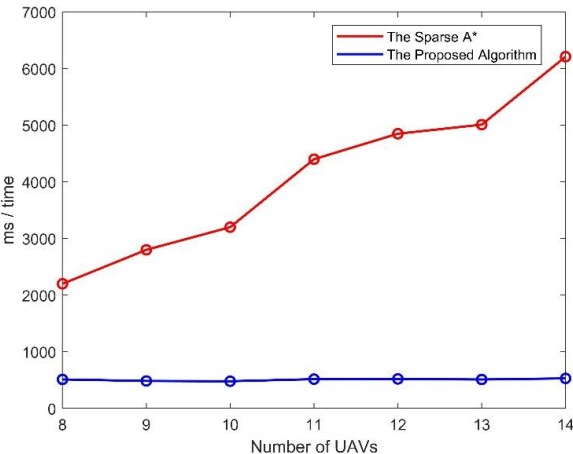

**Figure 24.** The algorithm running time.

We concluded from Figures 16 and 24 that the sparse A* algorithm was basically the same as the obstacle avoidance path planned by the algorithm in this paper, but the running time of the sparse A* algorithm was much longer than that of the algorithm in this paper, and it doubled with the increase in the number of UAV formations. However, the obstacle avoidance algorithm proposed in this paper based on geometry had a shorter running time and did not change significantly with the increase in the number of UAVs. It could meet online real-time trajectory planning, and has strong engineering applicability.

## 7. Conclusions

This paper proposed a method of fixed-wing UAV swarm formation control based on distributed ad hoc networks. This method included the formation switching of UAVs, formation obstacle avoidance, and handling of abnormal conditions during flight.

Simulation results showed that in formation switching, the ad hoc network, high-dynamics, fixed-wing UAV could form formation switching only by position information. The method was not sensitive to the initial position information of the UAV, which could eliminate errors during flight, and handle temporarily interrupted communication topologies and UAV drop, as well as other abnormal flight situations. In formation obstacle avoidance, the UAVs could be clustered into multiple subformations to pass through the obstacle avoidance area, and could be reconstructed as required formation. The formation technology was designed based on the waypoints, which was versatile, simple, and reliable, and is easy to realize in engineering.

Compared with the classic consistent formation control algorithm and obstacle avoidance algorithm, it was shown that the formation technology method proposed in this paper had lower complexity and higher timeliness, and is suitable for online formation flying of highly dynamic UAVs in an ad hoc network. However, the route planned by the method did not consider optimality, and it was only suitable for high-speed and constant-speed formation missions. For obstacle avoidance algorithms, the obstacle model is too simple, and there may be scenarios for obstacle avoidance that are not covered by the algorithm. Next, we will consider the optimal route planning problem and extend the algorithm to three-dimensional flight scenes to increase its practicability.

## 8. Patents

Suo, W.B., Zhang, D., and Wang, M.Y. "A distributed unmanned aerial vehicle flying around formation method based on time consistency", C.N. Patent, 202010226440.4, issued 10 July 2020.

Zhang, D., Suo, W.B., and Wang, M.Y. "A distributed unmanned aerial vehicle dynamic formation switching method based on waypoints", C.N. Patent, 202010226439.1, issued 10 July 2020.

**Author Contributions:** Conceptualization, W.S. and D.Z.; methodology, W.S.; software, W.S.; validation, W.S.; formal analysis, W.S.; investigation, W.S. and D.Z.; resources, D.Z. and Z.Q.; data curation, W.S., M.W. and L.Y.; writing—original draft preparation, W.S.; writing—review and editing, W.S.; visualization, W.S.; supervision, D.Z.; project administration, Z.Q.; funding acquisition, Z.Q. All authors have read and agreed to the published version of the manuscript.

**Funding:** This research received no external funding.

**Institutional Review Board Statement:** Not applicable.

**Informed Consent Statement:** Not applicable.

**Data Availability Statement:** The data presented in this study are available upon request from the corresponding author.

**Conflicts of Interest:** The authors declare no conflict of interest.

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
