# Peer review of "Formation Control Technology of Fixed-Wing UAV Swarm Based on Distributed Ad Hoc Network"

_applsci, doi:10.3390/app12020535_

Round 1

Reviewer 1 Report

This paper proposes a new swarm formation control method based on distributed ad hoc networks. Compared with rotary-wing UAVs, research on the formation of fixed-wing UAVs swarms has received less attention. 

In order to improve the paper quality, the following aspects should be clarified:
The proposed solution only applies to fixed-wing UAVs, or can it also be applied to other types of UAVs? Please justify.
The solution requires connectivity between all the UAVs. In ad hoc networks, this requirement is hard to ensure, and therefore, multi-hop communications are also used when direct communication is impossible. Therefore, the proposed solution can also be used in multi-hop network infrastructures?
The authors must provide more details about the simulation environment. 
The proposed method is never compared with a similar proposals; this is critical for any publication. 

Please update the reference sources whenever the sentence "Error! Reference source not found" occurs. 

Reviewer 2 Report

The article is very interesting but there are a few points to be made:
1. please elaborate more on the formulation of the problem with emphasis on the efficiency of the developed algorithm. please give a brief note on what can be improved in it and what does not work properly in it.
2. in line 230 there is a statement "Error! Reference source not found", please check it and correct it.
3. Please elaborate on the algorithm shown in Figure 5.
4. The mentioned error "Error! Reference source not found" error appears in the rest of the article, please correct it

5.  Please elaborate on the results obtained in the simulation study.
6. Please add more information about the effectiveness of the developed algorithm in the conclusions.
